# Pygo-F773W Mutation Reveals Novel Functions beyond Wnt Signaling in *Drosophila*

**DOI:** 10.3390/ijms25115998

**Published:** 2024-05-30

**Authors:** Youfeng Li, Zhigang Jiang, Yue Xu, Jing Yan, Qiong Wu, Sirui Huang, Lingxiao Wang, Yulian Xie, Xiushan Wu, Yuequn Wang, Yongqing Li, Xiongwei Fan, Fang Li, Wuzhou Yuan

**Affiliations:** The Laboratory of Heart Development Research, College of Life Science, Hunan Normal University, Changsha 410081, China; 202010140215@hunnu.edu.cn (Y.L.); 201201140149@hunnu.edu.cn (Z.J.); fan_xiongwei@163.com (X.F.)

**Keywords:** Pygo, Wingless (Wg)/Wnt signaling pathway, CRISPR/Cas9, RNA-seq, point mutation, posttranslational modification

## Abstract

Pygopus (Pygo) has been identified as a specific nuclear co-activator of the canonical Wingless (Wg)/Wnt signaling pathway in *Drosophila melanogaster*. Pygo proteins consist of two conserved domains: an N-terminal homologous domain (NHD) and a C-terminal plant homologous domain (PHD). The PHD’s ability to bind to di- and trimethylated lysine 4 of histone H3 (H3K4me2/3) appears to be independent of Wnt signaling. There is ongoing debate regarding the significance of Pygo’s histone-binding capacity. *Drosophila* Pygo orthologs have a tryptophan (W) > phenylalanine (F) substitution in their histone pocket-divider compared to vertebrates, leading to reduced histone affinity. In this research, we utilized CRISPR/Cas9 technology to introduce the Pygo-F773W point mutation in *Drosophila*, successfully establishing a viable homozygous *Pygo* mutant line for the first time. Adult mutant flies displayed noticeable abnormalities in reproduction, locomotion, heart function, and lifespan. RNA-seq and cluster analysis indicated that the mutation primarily affected pathways related to immunity, metabolism, and posttranslational modification in adult flies rather than the Wnt signaling pathway. Additionally, a reduction in H3K9 acetylation levels during the embryonic stage was observed in the mutant strains. These findings support the notion that Pygo plays a wider role in chromatin remodeling, with its involvement in Wnt signaling representing only a specific aspect of its chromatin-related functions.

## 1. Introduction

In 2002, four articles nearly simultaneously reported the discovery of a novel component of the Wnt signaling pathway, *Drosophila* Pygopus (dPygo), through various genetic screening approaches [1,2,3,4]. Currently, two proposed mechanisms regarding how Pygo participates in the Wnt signaling pathway have emerged. The first proposed mechanism suggests that during Wnt signaling pathway activation, Pygo is recruited through Lgs-Armadillo to *Drosophila* TCF (dTCF) target genes and subsequently recruits other transcription cofactors [5]. The second hypothesis is that, based on its histone-binding ability, Pygo functions as a de-repressor, alleviating Groucho-mediated inhibition at the initiation of Wnt signaling, thereby enabling the recruitment of Armadillo to dTCF target genes via Lgs [6]. However, it has also been reported that Pygo’s function is not always dependent on Wnt signaling. For example, Pygo was found to be involved in regulating the development of the eyes, teeth, brain, and intestine [7,8,9,10], as well as spermatogenesis [11], in a Wnt-independent manner.

In *Drosophila*, there is only one *Pygo* gene, while vertebrates harbor two *Pygo* homologs, *Pygo1* and *Pygo2*, both containing two highly conserved domains, NHD and PHD. The Pygo NHD is thought to recruit some transcriptional regulators and is essential for Wnt-regulated transcription [12]. The Pygo PHD is typically recognized to interact with Lgs/BCL9, consequently facilitating β-catenin binding [13,14]. However, while PHD proteins have long been implicated in the regulation of chromatin structure and function [15], the specific role of Pygo PHD in chromatin remodeling during Wg/Wnt signaling has not been fully elucidated. The PHD of Pygo proteins is capable of directly binding to di- and trimethylated lysine 4 of histone H3 (H3K4me2/3) [16], but research suggests that this interaction is dispensable for Wnt signaling-dependent transcription [17]. A reduction in Pygo2 expression levels leads to spermiogenesis arrest and consequent infertility through a specific decrease in lysine (K) 9/14 acetylation of histone H3, unrelated to Wnt signaling [11]. In human mammary epithelial cells, Pygo2 directly occupies the promoters of multiple histone genes and enhances the acetylation of lysine 56 in histone H3 (H3K56Ac) [18]. Additionally, Pygo2 can associate with MLL2 histone methyltransferase and Gcn5 histone acetyltransferase complexes through its first 47 amino acids [19]. Therefore, Pygo, in addition to its role in Wnt signaling, may also participate in posttranslational histone modification through its histone-binding capability.

The PHD fingers of vertebrate Pygo robustly interact with a histone H3 peptide methylated at lysine 4. However, in fruit flies, due to a single amino acid change (F773W), tryptophan is substituted with phenylalanine in their histone pocket-divider, which reduces their affinity for histones [20,21]. Research has reported that the PHD domain is utilized to link *Drosophila* Pygo to Legless/β-catenin, rather than for binding to histone H3, and this binding ability is not required for Pygo function in *Drosophila* [20]. Yet, another study concluded that an intact histone-binding pocket is critical for Pygo function during *Drosophila* development [22]. Furthermore, it was discovered that overexpression of humanized fly Pygo specifically in the *Drosophila* wing disc not only hyperactivates Wnt signaling components but also derepresses Notch targets [21]. In mice, mutations leading to reduced levels of Pygo2 or abolishing the binding ability of Pygo2 to H3K4me2/3 result in male infertility. Moreover, mechanistic studies indicate that during spermiogenesis in the testis, Pygo2 may not operate via Wnt signaling but instead could potentially function through the HAT enzyme Gcn5 to facilitate H3 acetylation. However, unlike in mice, abrogating the ability of dPygo to bind histones in fruit flies does not result in reduced fecundity [11,17]. Therefore, the question regarding the PHD’s binding capability to histone H3 remains a subject of controversy and may be a breakthrough point in studying Pygo’s functions beyond the Wnt pathway.

All previously obtained *Drosophila Pygo* mutants generated through various mutagenesis methods have displayed homozygous lethality. Additionally, knockdown or overexpression of Pygo throughout the entire body also resulted in failure to develop into adult fruit flies. Moreover, Pygo exhibits dose effects and significant maternal effects. Early research primarily relied on tissue-specific mutation or overexpression to explore the functionality of *Drosophila Pygo*, which may pose limitations. In this study, we utilized CRISPR/Cas9 technology to create a humanized fly *Pygo* mutation (Pygo-F773W), which enhances the binding capability of *Drosophila* Pygo to histone H3. We successfully obtained a viable homozygous *Pygo^F773W/F773W^* (described below as Pygo-F773W) mutant strain for the first time. The mutant adult flies displayed noticeable abnormalities in reproduction, locomotion, heart function, and lifespan. RNA-seq and cluster analysis unveiled that the mutation does not primarily disrupt the Wnt signaling pathway. Instead, it predominantly impacts pathways related to immunity, metabolism, and posttranslational modification in adult flies. Possibly due to Pygo’s spatiotemporal and tissue-specific nature, no differences in H3K9 acetylation levels were observed in the mutant adult fruit flies, while a decrease in H3K9 acetylation levels during the embryonic stage was noted. Hence, our findings offer valuable guidance for further exploration of the Wnt-dependent or Wnt-independent mechanisms of Pygo.

## 2. Results

### 2.1. Construction of Pygo-F773W Point Mutation Drosophila Line

With the development of gene-editing technology, we can utilize CRISPR/Cas9 technology [23] to efficiently introduce mutations in the target gene. *Pygo* containing the F773W mutation was created through homologous recombination-mediated knock-in using the CRISPR/Cas9 system. We designed a pair of small guide RNAs (sgRNAs), namely Target 1 and Target 2 (Figure 1A,B), along with the donor template for homologous recombination. The mutation site in the homology-directed repair (HDR) template contains the point mutations and a few synonymous mutations designed to increase the efficiency of homology-directed repair of the donor (Figure 1B). We obtained a viable homozygous Pygo-F773W mutant *Drosophila* line and verified it via Sanger sequencing (Figure 1C).

### 2.2. Mutant Male and Female Drosophila Display Impaired Fertility

In the Pygo-F773W mutant *Drosophila* homozygous line, we observed a significant reduction in the number of offspring compared to the wild type (*w^1118^*). In the fertility experiment, a single male was crossed with a single virgin female per tube, and the count of fertile flies and viable offspring per female was recorded over a 48-h period. It was observed that nearly all hybridizations involving wild-type male flies resulted in viable progeny when paired with either wild-type or Pygo-F773W mutant females. However, only one-third of hybridizations involving Pygo-F773W mutant male flies with either wild-type or Pygo-F773W mutant females led to viable progeny (Figure 2A). Furthermore, statistical analysis of the number of viable progenies per female within the 48-h timeframe revealed that Pygo-F773W mutant female flies produced significantly fewer viable offspring compared to wild-type females (Figure 2B). These findings indicate a reduced fecundity in both male and female mutant flies in comparison to wild-type flies.

### 2.3. Mutant Flies Are Inactive and Have Poor Movement Ability

Subsequent experiments were carried out on the fruit flies, utilizing a behavior trajectory tracking system to examine the movement patterns of Pygo-F773W mutant flies in comparison to wild-type flies over a one-hour period. The findings revealed that the mutant flies displayed reduced activity levels and covered a shorter total distance within the designated timeframe (Figure 3A,B).

We next performed a climbing ability analysis to evaluate the locomotor behavior of each strain of flies. The tube climbing experiment is a classic method of measuring the movement ability of flies. After all the fruit flies were shaken to the bottom of the tube, the crawling time taken by the first four flies of each strain to climb to the 6 cm mark was recorded. A shorter climbing time indicates better movement ability in fruit flies. Compared with the *w^1118^* flies, both female and male Pygo-F773W mutant flies exhibited inferior climbing abilities (Figure 4A). Because of the abnormalities in *Drosophila*’s locomotor ability, we considered whether there were abnormalities in the muscle structure in *Drosophila*. The F-actin of dissected fruit flies was stained using fluorescently tagged phalloidin (Actin-Tracker Green-488). The results revealed no apparent abnormalities in the body wall muscles of the mutant flies (Figure 4B) compared to wild-type flies (Figure 4C). While Pygo-F773W may not directly impact muscle structure, it could potentially affect the movement of *Drosophila* through alternative pathways such as energy metabolism or ion exchange pathways. Additionally, the movement of *Drosophila* may also be influenced by abnormal neurodevelopment or heart function. These factors highlight the need for additional research. In summary, the Pygo-F773W mutation leads to reduced activity and locomotion in flies.

### 2.4. Abnormal Cardiac Function Induced by Pygo-F773W Mutation

Previous research in our laboratory revealed that Pygo plays a critical role in adult heart function and is Wnt signaling-independent [24]. We also investigated the impact of Pygo-F773W mutation on adult heart function through the use of a semi-automated optical heartbeat analysis (SOHA) [25,26,27]. Videos lasting 30 s were recorded at 110 frames per second using a high-speed electron-multiplying charge-coupled device (EM-CCD) camera. This technique allows for the analysis and quantification of the rhythmicity and the dynamics of heart contractions, including the heart period (HP), diastolic interval (DI), systolic interval (SI), diastolic diameter (DD), systolic diameter (SD), fractional shortening (FS, a classic measure of cardiac output), and arrhythmia index (AI) (Figure 5). The arrhythmia index (AI) refers to the difference between each heartbeat cycle of the *Drosophila* heart and the median value of the entire heartbeat cycle, representing the degree of irregular heartbeat in *Drosophila*.

We found that Pygo-F773W mutant hearts exhibited dramatic abnormalities across most of the cardiac parameters measured, which became more severe with age (Figure 6). Compared with wild-type *w^1118^* flies, the Pygo-F773W mutation prolonged the heart period in 1-week-old and 3-week-old flies (Figure 6A), and this was due to increases in both the systolic and diastolic intervals (Figure 6B,C). We also observed a significant increase in the incidence of arrhythmias in Pygo-F773W mutant hearts compared to controls (Figure 6D). The cardiac fractional shortening (FS) of the mutant *Drosophila melanogaster* was not significantly different from that of the wild types at 1-week-old, but it was significantly reduced at 3 weeks (Figure 6E). And, in 3-week-old Pygo-F773W mutant hearts, we noticed the occurrence of fibrillation (Figure 5B). Since most Pygo-F773W mutant flies died after 3 weeks, we were unable to perform a SOHA analysis on older flies. Overall, these data indicate that the Pygo-F773W mutation causes an age-dependent deterioration of heart function.

### 2.5. Pygo-F773W Mutation Reduces Egg-to-Adult Survival and Post-Eclosion Lifespan

Although adult fruit flies homozygous for the Pygo-F773W mutation were obtained, their viability was found to be significantly reduced. Subsequent analysis focused on the impact of the Pygo-F773W mutation on the *Drosophila* percent survival to adulthood and post-eclosion lifespan. In comparison to the *w^1118^* wildtype group, where 20 wildtype eggs yielded an average of 16.5 pupae and 16 adult flies, the same number of eggs with the Pygo-F773W mutation only produced an average of 12.5 pupae and 10.5 adult flies. Notably, the Pygo-F773W mutation led to considerable lethality during both the egg-to-pupae and pupae-to-adult fly transitions (Figure 7A). To further investigate the lifespan of mutant flies in comparison to wildtype flies, a survival graph was plotted, and a log-rank (Mantel–Cox test) analysis was performed using GraphPad Prism. The median survival for female flies with the Pygo-F773W mutation was 12 days, which was significantly shorter than the 57 days observed for the control group, while male flies with the mutation survived for a median of 15 days compared to 51 days for the control (Figure 7B). Taken together, these findings indicate that the Pygo-F773W mutation not only reduces the survival rate from egg to adult fly but also significantly shortens the post-eclosion lifespan.

### 2.6. Transcriptome Sequencing Data Analysis

*Drosophila* Pygo can participate in regulating TCF target genes’ transcriptional activation through the Wnt signaling pathway. Similarly, Pygo binds to histone H3 to regulate its methylation or acetylation; therefore, it may also contribute to transcriptional regulation. We explored the effects of the Pygo-F773W mutation on gene expression using high-throughput transcriptome sequencing. Compared with the wild-type *w^1118^* group, the Pygo-F773W group exhibited 351 upregulated and 126 downregulated differentially expressed genes (DEGs) (Figure 8A, Appendix A). The heatmaps of the DEGs are shown in Figure 8B. A GO (gene ontology) analysis revealed that their roles were mainly involved in proteolysis, the innate immune response, carbohydrate metabolic processes, extracellular space, oxidoreductase activity, and iron ion binding (Figure 9A). A KEGG (Kyoto Encyclopedia of Genes and Genomes) pathway enrichment analysis revealed that the Pygo-F773W mutation mainly affects drug metabolism, lysosome, toll and lmd signaling pathways, and longevity-regulating pathways (Figure 9B).

### 2.7. Wnt Signaling Pathway Is Not the Major Pathway Affected by Pygo-F773W Mutation

Pygo has been identified as a specialized nuclear co-activator that is specifically involved in the canonical Wnt signaling pathway. However, the KEGG pathway enrichment analysis of DEGs did not show that the Wnt signaling pathway was significantly enriched. The Wnt pathway annotation results of DEGs showed that there were no significant changes in the key genes of the Wnt pathway. Only the upregulation of the TAK1 and the downregulation of the NLK related to the MAPK signaling pathway may have affected the expression of Wnt downstream target genes (Figure 10A). General differential analysis usually only focuses on some significantly upregulated or downregulated genes, which will miss some genes that are not significantly differentially expressed but have important biological significance. We further performed a gene set enrichment analysis (GSEA) on the transcriptome results. A GSEA does not need to specify a clear threshold for differential genes. It sorts all genes according to their differential expression levels into two sets of samples and can detect weak but consistent trends. The top five pathways with the most reliable significance (i.e., the smallest *p*-value) in the enrichment analysis were single-organism metabolic process, primary metabolic process, the defense response to bacterium, cuticle development, and sulfur compound metabolic processes (Appendix A). Meanwhile, we also found downregulation of Wnt signaling pathway via detailed analysis of the GSEA results (Figure 10B). Therefore, these results indicate that while the Pygo-F773W mutation influences the Wnt signaling pathway, it is not the primary target of this mutation.

### 2.8. Tissue-Specific Enrichment Analysis and PPI Network Construction of DEGs

It has been previously reported that the organs where Pygo acts in a Wnt independent manner mainly include the eyes, teeth, testis, brain, intestine, and salivary gland [7,8,9,10,11]. To explore the sources of these altered genes in Pygo-F773W mutant flies, the DEGs were submitted to Metascape for tissue-specific enrichment analysis using the PaGenBase database, a database repository for the collection of tissue- and time-specific pattern genes. We found that the DEGs were mainly located in the midgut, head, epidermis with attached muscle, hindgut, and salivary gland, which was consistent with the above reported organs, indirectly suggesting that the Pygo-F773W mutation may function in a Wnt-independent manner (Figure 11A). Subsequently, we also conducted core gene screening of the DEGs. First, the STRING database was employed to analyze the network diagram of the DEGs. Then, using Cytoscape software (Version 3.10.1), the connectivity (degree) was organized from high to low, producing a precise PPI regulatory network diagram. The PPI network diagram only showed genes with a degree greater than 10. Genes with degrees greater than 20 and 25 in the PPI network correspond with hub genes and key genes, respectively (Figure 11B). Then, 18 hub genes and 5 key genes were identified, and their expression patterns were visualized using a heatmap based on RNA-seq data (Figure 11C).

### 2.9. Pygo-F773W Mutation Related to Posttranslational Modification

The Cluster of Orthologous Groups of Proteins (COG) is a database that categorizes proteins into orthologous groups. The proteins constituting each type of COG are assumed to be derived from an ancestor protein and therefore are either orthologs or paralogs. Posttranslational modification, protein turnover, and chaperones contained the greatest number of DEGs (Figure 12A), including key gene Jon99Ci and hub genes Jon44E, Jon25Bii, Jon66Cii, CG18180. A PPI network analysis of the DEGs related to posttranslational modification was conducted using STRING online tools (Figure 12B). Additionally, a KEGG pathway enrichment analysis was performed using DAVID functional annotation (Figure 12C). The KEGG pathway analysis revealed that the mutations mainly affect longevity-regulating pathways, endocytosis, protein processing in endoplasmic reticulum, spliceosome, and neuroactive ligand–receptor interactions. These results indicate that the Pygo-F773W mutation may mainly affect posttranslational modification.

### 2.10. Pygo-F773W Mutation Inhibits Acetylation of Histone H3 Lysine-9 (H3K9) in Drosophila Embryos

Posttranslational histone modification plays an important role in development. Pygo was detected to be physically associated with the Gcn5-containing STAGA HAT complex. Research has demonstrated that Pygo2 recruits the histone acetyltransferase (HAT) enzyme Gcn5 to chromatin, facilitating the acetylation of histone H3 during spermiogenesis in mice independently of Wnt signaling [11,17]. Gcn5 is known to acetylate lysine 9 and lysine 14 on histone H3. Consequently, it was hypothesized that the Pygo-F773W mutation in *Drosophila*, which enhances Pygo’s binding to histone H3, might lead to increased levels of histone H3 acetylation. However, the experimental results did not support this hypothesis. A Western blot analysis of histone H3 lysine 9 acetylation (H3K9-ac) in adult *Drosophila* did not show significant changes in the acetylation level. Surprisingly, there was a decrease in H3K9 acetylation in *Drosophila* embryos (Figure 13). This unexpected finding suggests that the Pygo-F773W mutation may have differential effects on histone acetylation in different developmental stages or tissue types in *Drosophila*. Further investigation is required to elucidate the underlying mechanisms and implications of these results.

## 3. Discussion

The mechanism by which Pygo operates in a Wnt-dependent or Wnt-independent manner is still not fully understood. The focus of research centers around the histone-binding capability of the Pygo PHD domain. Some studies propose that the PHD domain primarily interacts with Legless/β-catenin and may not require histone-binding functionality. However, other research suggests that histone-binding ability is indeed crucial for the PHD domain of the Pygo [17,20,21,22]. The histone-binding ability of Pygo differs between *Drosophila* and mammals due to a single amino acid difference. Compared with mammals, *Drosophila* Pygo orthologs exhibit a tryptophan (W) > phenylalanine (F) substitution in their histone pocket-divider, which reduces their affinity for histones. In mammals, Pygo has evolved two homologous genes, Pygo1 and Pygo2. While the histone-binding residues are the same between the two human Pygo paralogs, a study already shows that Pygo2 complexes exhibit slightly higher binding affinities for methylated histone H3 tail peptides than Pygo1 complexes [28]. It is possible that mammalian Pygo1 and Pygo2 function together in a manner similar to Pygo in *Drosophila*.

Prior to this, various Pygo mutant strains were screened, including *pygo^F66^*, *pygo^F15^*, *pygo^F107^*, *pygo^10^*, *pygo^9^*, *S123*, and *S28* [2,3,4], all of which exhibited homozygous lethality and showed a high degree of maternal effect. Additionally, Pygo also displayed dosage effects. Pygo is a crucial factor in the Wnt signaling pathway; however, overexpression of Pygo has been found to inhibit Wnt signaling. Dosage effects have also been observed in mouse experiments [11]. These factors may greatly interfere with and limit subsequent research on Pygo. In this study, we employed CRISPR/Cas9 technology to introduce the site-specific mutation Pygo-F773W in *Drosophila*, and we obtained a viable homozygous Pygo mutant line. The mutant adult flies displayed conspicuous abnormalities, including a reduced reproductive capacity of both male and female fruit flies, relative inactivity and poor exercise capacity, age-dependent cardiac dysfunction, and a highly significantly shortened lifespan. However, no obvious abnormalities were observed during embryonic development in *Drosophila* carrying the Pygo-F773W mutation. Their cuticles and denticles appeared normal, lacking the hallmark characteristics of *Drosophila* Wnt signaling abnormalities, which show no naked cuticle and have a characteristic denticle lawn phenotype [29,30]. Pygo-F773W may not mainly affect Wnt signaling, which is also reflected in the transcriptome data. And, early studies in our laboratory found that specific knockdown of Pygo in the adult *Drosophila* heart leads to abnormal cardiac function in a Wnt signaling-independent manner, which is very similar to the phenotype caused by the Pygo-F773W mutation. We have also explored how whole-body knockdown of Drosophila Pygo leads to developmental delays and hindered eclosion. A transcriptome analysis of surviving larvae revealed a significant impact on the Wnt signaling pathway following Pygo whole-body knockdown. Therefore, we speculate that the Pygo-F773W mutation may have minimal impact on the embryonic development stage of *Drosophila* and does not perturb the Wnt pathway significantly. Instead, its primary function likely manifests during the adult stage, particularly in the gut and head of *Drosophila*.

A transcriptome analysis revealed that the Pygo-F773W mutation primarily regulates posttranslational modification, with related genes falling into two main categories. The first category comprises the S1A serine proteases (SPs), such as Jon99Ci, Jon99Cii, Jon99Ciii, Jon25Bii, Jon44E, Jon66Cii, Jon74E, CG30002, CG18180, CG18179, and SPH93. These serine proteases play crucial roles in various physiological processes, including the digestion of dietary proteins, blood coagulation, complement activation, fertilization, and tumor metastasis in mammals [31,32,33]. The second category includes heat shock proteins (HSPs), such as Hsp68, Hsp70Ab, Hsp70Bb, Hsp70Bbb, Hsp70Bc, and Hsp40 (DnaJ-1), which are molecular chaperones that play crucial roles in various biological processes. The main functions of HSPs encompass molecular chaperones, resistance to stress, immune response, and inhibition of apoptosis [34,35,36]. The 90 kDa heat shock protein (HSP90) is a therapeutic target for cardiovascular diseases (CVDs) and heart aging [37]. The 70 kDa heat shock protein (HSP70) may maintain the normal structure and function of cardiomyocytes by participating in myocardial ion channel repair, preventing protein denaturation and stabilizing the cell membrane and lysosomal membrane. Ectopic expression of HSP70 in rat hearts increased the tolerance of the heart to ischemic injury [38]. Increased levels of HSP60 in cardiomyocytes may provide protection through an indirect anti-apoptotic effect. HSP27 could potentially safeguard the myocardium by preventing cell lysis [39]. Research has also demonstrated that HSP70 plays a crucial role in protecting the brain from various nervous system injuries [40]. While different HSPs have distinct protective functions in heart and nervous system disease, the specific mechanisms underlying their effects remain unclear, warranting further investigation. Furthermore, the core genes in differentially expressed genes (DEGs) include two other types of genes: α-glucosidase (Mal-A1, Mal-A3, Mal-A4, Mal-A8) and Bomanins (BomS1, BomS3, BomBc1). Bomanins, which are small, secreted peptides induced by Toll signaling, are part of a family of proteins that may provide resistance to bacterial infections [41,42].

Additionally, Pygo itself may also be involved in posttranslational histone modification. We found that there was a decrease in H3K9 acetylation in Pygo-F773W-mutant *Drosophila* embryos, but no change in H3K9 acetylation was detected in adults. This discrepancy could be attributed to the tissue-specific or spatio-temporal nature of Pygo function, leading to reduced acetylation levels in adult *Drosophila* that may not be detectable in the entire body of the organism. The Pygo-F773W mutation was expected to enhance Pygo’s binding to histones, leading to an increase in histone acetylation. Surprisingly, the mutation resulted in a decrease in histone acetylation levels. Our speculation is that Pygo may competitively bind H3K4me2/3 with acetylase or methylase while recruiting them in *Drosophila*. Further comprehensive studies are needed to explore these mechanisms.

In summary, we engineered a Pygo-F773W point mutation in *Drosophila* using CRISPR/Cas9 technology. Homozygous *Drosophila* carrying the mutation survive but display notable abnormal phenotypes. A transcriptome analysis indicated that the mutation did not primarily impact the Wnt pathway, but instead, it predominantly modulated adult *Drosophila* development through the posttranslational modification pathway (Figure 14). These findings offer valuable insights and potential avenues for further in-depth investigations into Pygo.

## 4. Materials and Methods

### 4.1. Drosophila Husbandry

*Drosophila* stocks were cultured in standard media at 25 °C with 60% humidity in a 12 h light and 12 h dark cycle. When collecting virgin flies and performing other experiments, the fruit flies were anesthetized by passing a small stream of carbon dioxide directly into the culture tube and then transferring it to a plate where carbon dioxide was continuously flowing.

### 4.2. CRISPR/Cas9-Mediated Pygo-F773W Point Mutation

CRISPR/Cas9 mutagenesis was performed as previously described [23,43,44].Two sgRNA (the gRNA sequences are listed in the Results section) plasmids for the Pygo gene and donor plasmid were injected into fly embryos (*w^1118^*). The mutation site in the donor sequence contained the point mutations and a few same-sense mutations which were designed to increase the efficiency of the homology-directed repair of the donor. When the injected P0 embryos grew into adults, they were crossed with TM2/TM6B. The genomic DNA of the P0 and F1 flies were extracted. PCR was performed using primers for validation of the point mutation. The F2 flies from positive F1 tubes were balanced with TM6B. The validation primers were as follows:

CG11518-F773W-1F:TATGGGCGTAGGTCCCAAGCCGA

CG11518-F773W-3R:GCTTTTAGTTCTGAGATCAGTAG.

### 4.3. Semi-Intact Drosophila Heart Preparation and Cardiac Function Analysis

Semi-intact *Drosophila* hearts were prepared as described previously [45,46]. Movies of beating hearts were recorded for 30 s using a high-speed EM-CCD camera (Hamamatsu; Shizuoka; Japan 100–140 fps/s) at 110 frames/s. Data was captured using HCImage Imaging Software (Hamamatsu, Version 2.1.1.0). The movies were analyzed using with Semi-Automatic Optical Heartbeat Analysis software (SOHA, Version 3.4.0.0, provided by Ocorr and Bodmer) to quantify heart periods, systolic and diastolic intervals, systolic and diastolic diameters, fractional shortening, and arrhythmia indexes and to produce M-mode records.

### 4.4. Fertility Test

*w^1118^* and Pygo-F773W mutation virgin flies were collected every 4 h to prevent hybridization. Male flies of the same period were collected into another empty culture tube. Then crosses of 2-day-old single males and 2-day-old single virgin females were set up in separate tubes, the number of flies having progeny was counted, as well as the number of viable progeny per female obtained in a 48 h time period. In order to ensure that the fruit flies in each tube could mate successfully, we chose a long hybridization time of 48 h and ensured that both male and female fruit flies were still alive after 48 h.

### 4.5. Lifespan Assay

Adult male and female flies were collected on the day of eclosion and maintained with 25 flies per tube at 25 °C with 60% humidity and a 12 h light/12 h dark cycle. The flies were transferred to new tubes every 3 days and scored for survival. A log-rank (Mantel–Cox) test was used to compare the lifespans among the different genotypes. Two replicates (about 200 flies in each genotype in every replicate) were used in each test.

### 4.6. Phalloidin Staining

Semi-intact *Drosophila* abdominal muscles were fixed in situ using 4% paraformaldehyde/PBS for 20 min. Then the abdominal muscles were washed three times in PBST (0.1% Triton X-100/PBS). The muscles were stained with Actin-Tracker Green-488 (Beyotime, C2201S) for 2 h and washed three times with PBS at room temperature (10 min for each wash). Fluorescence staining images were obtained using a fluorescence microscope (Leica M205FA, Wetzlar, Germany).

### 4.7. Behavioral Trajectory Tracking Analysis

After using carbon dioxide to anesthetize the fruit flies, the fruit flies were quickly transferred to a 48-well petri dish. One fruit fly was placed in each well, and the lid was closed. The dish was left at room temperature for 2 h to ensure that the fruit flies fully recovered from anesthesia. Then, the dish was placed into a zebrafish behavioral observation box (France/ViewPoint/ZebraBox). The “ViewPoint Application Manager” behavior analysis software(Version 5.18.0.0) was opened, and the “behavior tracking” experiment type was selected. The thresholds for low, medium, and high speeds were set based on the speed of fruit fly movement: low-speed movement (<1 cm/s); medium-speed movement (>1 cm/s and <2 cm/s); high-speed movement (>2 cm/s). The movement of the fruit flies was monitored in real time within 1 h, and we obtained the movement trajectory diagram within 1 h and the total distance moved at each speed for subsequent analysis.

### 4.8. RNA Sequencing and Data Analysis

RNA sequencing was performed by Tsingke. After RNA extraction, purification, and library construction, the samples underwent next-generation sequencing (NGS) using the Illumina sequencing platform. The libraries were subjected to paired-end (PE) sequencing. Three replicates (five male and five female whole flies in each replicate) were used for the *w^1118^* and Pygo-F773W groups at 3 days of age. DEGs were analyzed using DESeq with screening conditions set at a significance threshold of |log2FC|>1 and *p* < 0.05. GO and KEGG pathway enrichment analyses were performed using David v2022 (https://david.ncifcrf.gov/, accessed on 24 March 2024), with *p* < 0.05 used as the cut-of criteria. PPI networks were constructed using STRING version 11.0b (https://string-db.org/, accessed on 25 March 2024) and Cytoscape (version 3.10.1). DEGs were sorted based on degree values, where degrees greater than 20 and 25 in the PPI network corresponded with hub genes and key genes, respectively. Heat maps were plotted using the website (https://www.bioinformatics.com.cn, accessed on 25 March 2024), a free online platform for data analysis and visualization.

### 4.9. Western Blot

Total protein extracts were prepared according to standard protocols, and proteins were subjected to SDS-PAGE separation and blotting. The following antibodies were used: acetyl-histone H3 (Lys9) rabbit polyclonal antibody (Beyotime, Shanghai, China, AF5611, 1:1000 dilution); histone H4 rabbit polyclonal antibody (Beyotime, Shanghai, China, AF7107, 1:1000 dilution); antibody to GAPDH (Sigma-Aldrich, Burlington, MA, USA,1:2000 dilution). Antibody binding was detected using an ChemiDocTMXRS+ system (Bio-rad, Shanghai, China).

### 4.10. Data Analysis and Visualization

Statistical analysis and graphing of the data was carried out using GraphPad Prism 8.0 Software. An unpaired *t*-test was used to assess the differences between groups. Data were presented as means ± SEM from at least 3 replicates, ns *p* > 0.05, * *p* < 0.05, ** *p* < 0.01, *** *p* < 0.001.

## Figures and Tables

**Figure 1 ijms-25-05998-f001:**
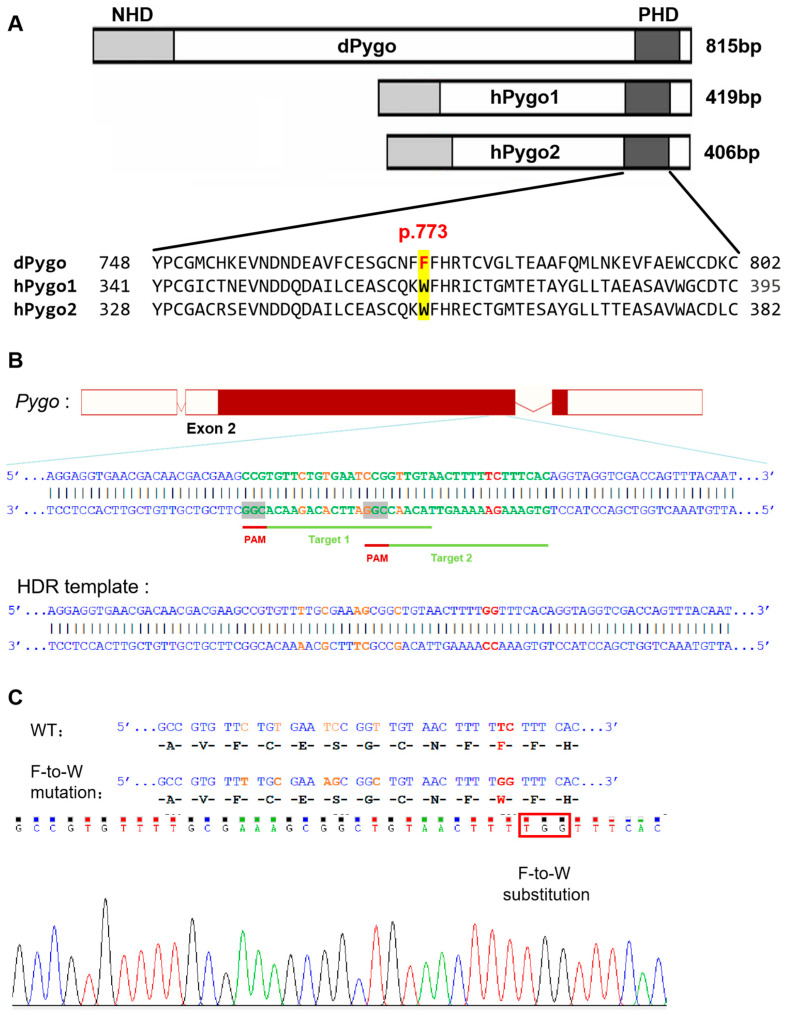
Schematic diagrams of Pygo-F773W point mutation in Drosophila. (**A**) Alignment of PHD finger sequences from Pygo orthologs of Drosophila and human. Yellow marks indicate phenylalanine at position 773 of Drosophila Pygo protein and its corresponding tryptophan in human Pygo1 and Pygo2 proteins. (**B**) **Upper panel**: Schematic diagram of sgRNA targeting for Pygo-F773W point mutation. The red horizontal line represents the genomic DNA of Pygo, rectangles represent the exons of Pygo, the red rectangle represents the coding sequence (CDS), the green font represents the target site sequence, and the red font represents the protospacer adjacent motif (PAM). **Lower panel**: Schematic diagram of a partial sequence of the homology-directed repair (HDR) template. Dark red marks indicate the bases of Pygo-F773W point mutations, and light red marks indicate the bases of same-sense mutations. (**C**) Sequencing peak diagram of mutant alleles, in which bases A, G, C, and T are represented by green, black, blue, and red curves, respectively.

**Figure 2 ijms-25-05998-f002:**
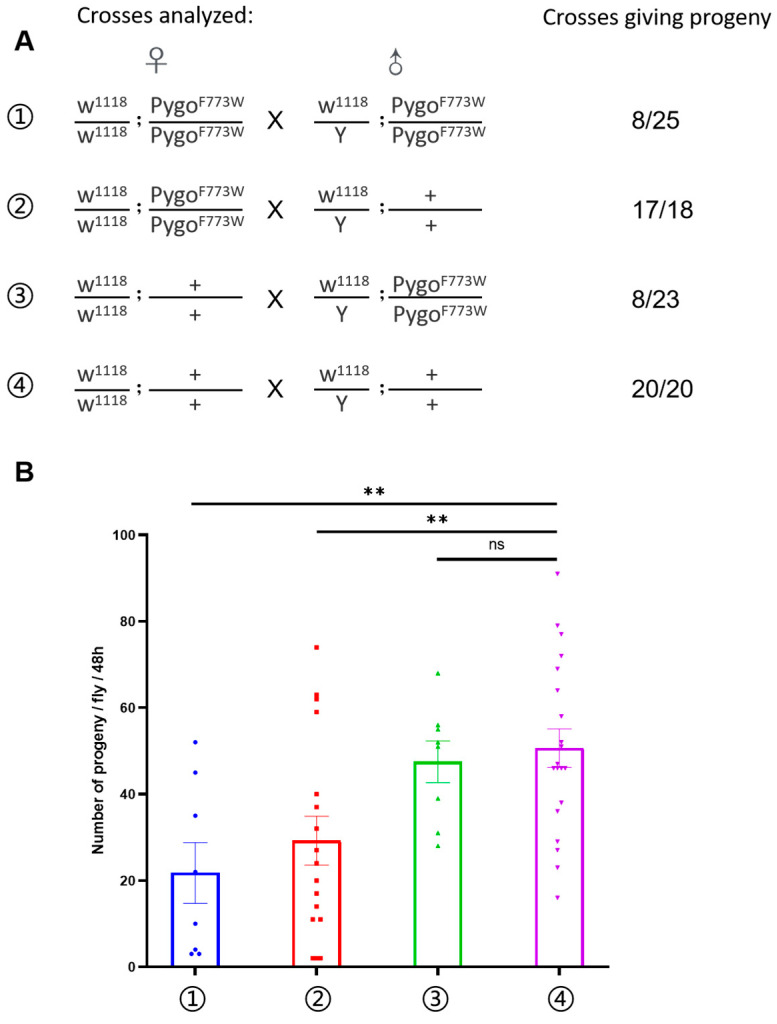
Pygo-F773W mutation impaired the fertility of *Drosophila melanogaster*. (**A**) Genotypes of flies used in the hybridization experiment. **Right**: the number of flies having progeny divided by the total number of flies tested for each hybridization. (**B**) The number of viable progeny per female obtained in a 48 h time interval. Plotted data represent means ± standard error of the mean (SEM). ns *p* > 0.05; ** *p* < 0.01.

**Figure 3 ijms-25-05998-f003:**
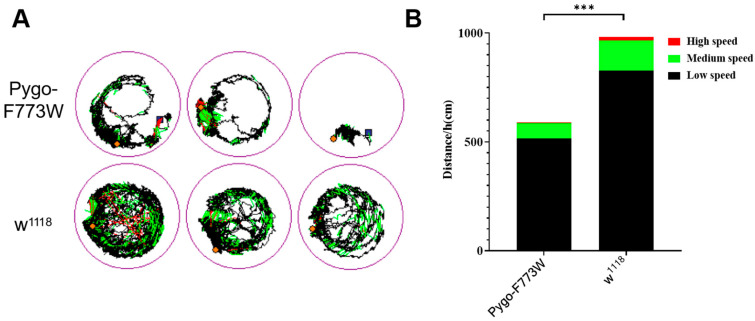
Movement trajectories and statistical analysis of movement distances. (**A**) Movement trajectories of 3-day-old Pygo-F773W mutant flies (up) and *w^1118^* flies (down). Black line: low-speed movement (<1 cm/s); green line: medium-speed movement (>1 cm/s and <2 cm/s); red line: high-speed movement (>2 cm/s). (**B**) Statistical analysis of the distance traveled by the fruit flies in 1 h at various speeds. *** *p* < 0.001 (n = 10, unpaired *t*-test).

**Figure 4 ijms-25-05998-f004:**
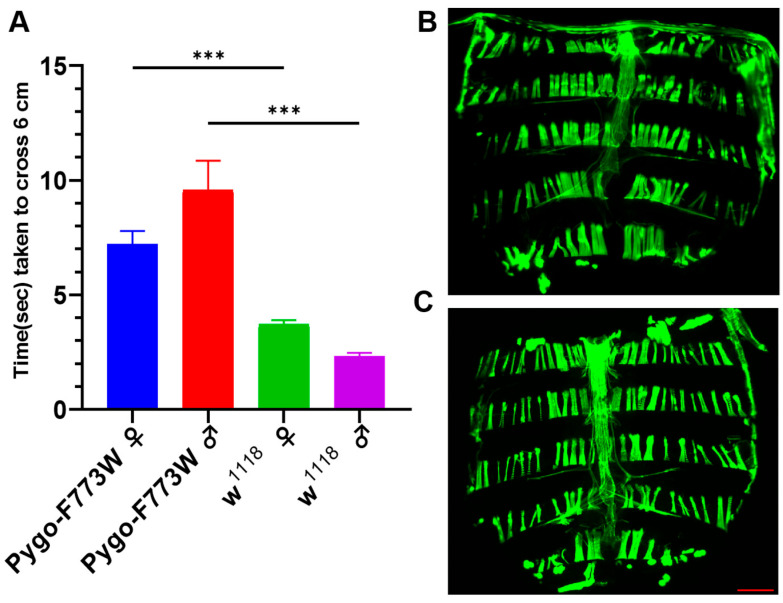
Climbing ability analysis and F-actin staining. (**A**) Climbing ability analysis of 3-day-old Pygo-F773W mutant flies compared to *w^1118^* flies, separately for males and females. (**B**,**C**) F-actin staining using fluorescently tagged phalloidin. Climbing sample n = 100. Plotted data represent means ± SEM, *** *p* < 0.001 (unpaired *t*-test); scale bar: 200 μm.

**Figure 5 ijms-25-05998-f005:**
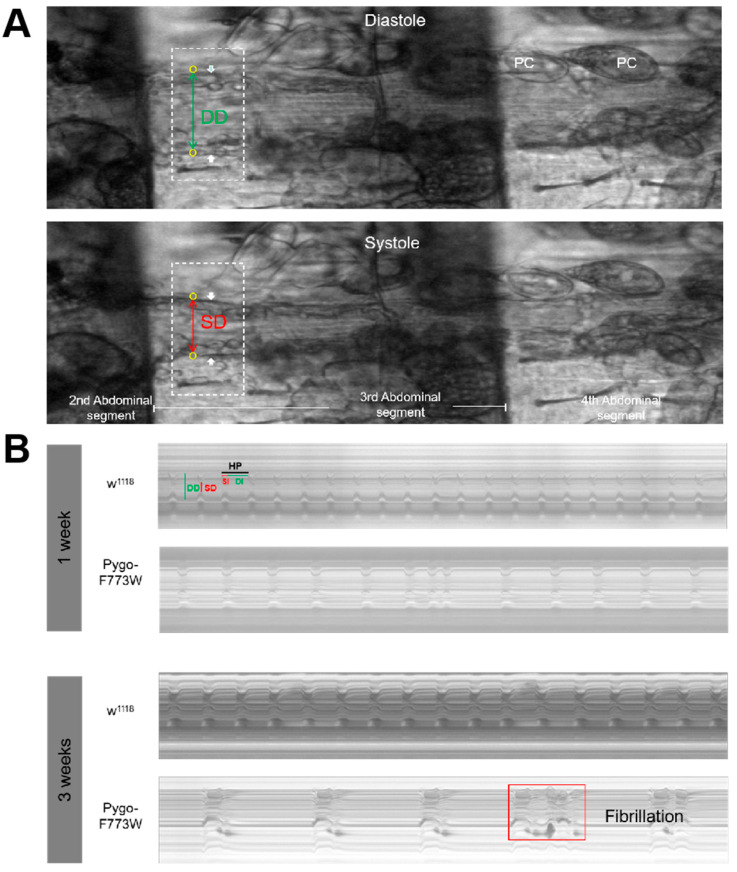
Movement detection from high-speed digital movies. (**A**) Still image of three parts of a fly’s abdomen showing the exposed heart during diastole and systole. The position of the heart wall is indicated by the white arrow, and the circles represent the positions where the procedure is marked and used to calculate the heart diameter. The dashed rectangle shows the orientation of pixel strips used to generate the M-modes shown in (**B**). (**B**) Representative 10 s M-mode traces from semi-intact *Drosophila* heart preparations reveal the movements of the heart walls (y-axis) over time (x-axis). 1-week-old and 3-week-old Pygo-F773W mutant flies are significantly slower and more irregular than *w^1118^* flies, and fibrillation appears at 3 weeks old. PC: pericardial cell; HP: heart period; SI: systolic interval; DI: diastolic interval; DD: diastolic diameter; SD: systolic diameter.

**Figure 6 ijms-25-05998-f006:**
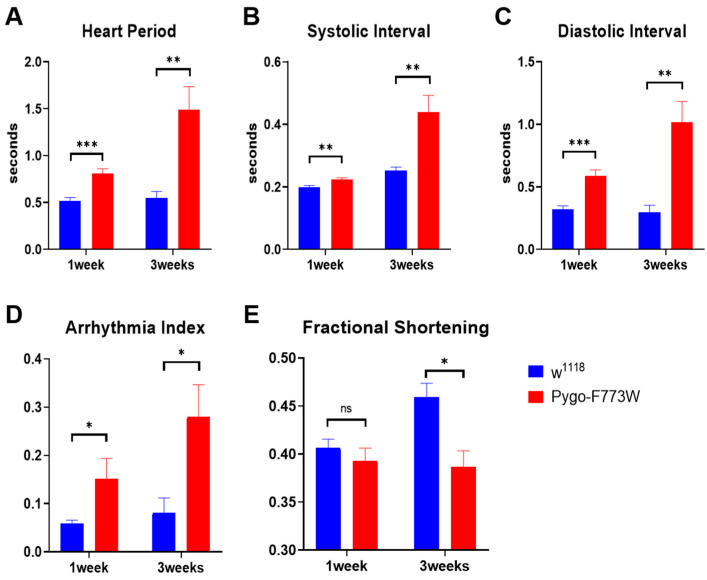
Effects of Pygo-F773W mutation on cardiac function with age. (**A**) Heart period, (**B**) systolic interval, (**C**) diastolic interval, and (**D**) arrhythmia index are increased in Pygo-F773W mutants compared to controls (*w^1118^*) and are further exacerbated with age. (**E**) Fractional shortening is not significantly reduced at 1 week but becomes significantly reduced at 3 weeks. For each data point (i.e., at 1 and 3 weeks of age), 10–30 individual hearts from flies per genotype were analyzed. Data are displayed as mean ± SEM; ns *p* > 0.05; * *p* < 0.05; ** *p* < 0.01; *** *p* < 0.001 (unpaired *t*-test).

**Figure 7 ijms-25-05998-f007:**
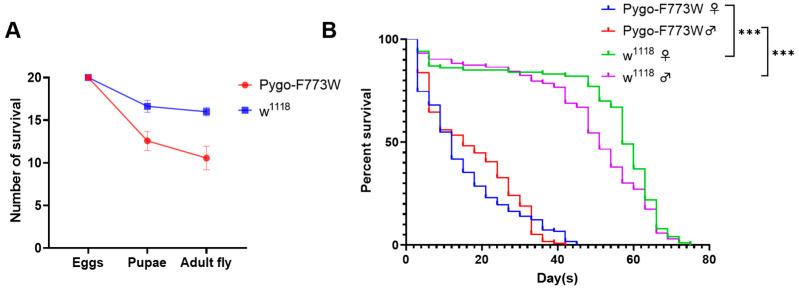
Percent survival from eggs to adult *Drosophila* and the lifespan of adult *Drosophila*. (**A**) The graph shows survival number of *Drosophila* at two different developmental stages when eggs were transferred to new medium. Data analysis was performed with 20 eggs per tube and repeated 8 times. Pygo-F773W mutant flies compared to *w^1118^* flies, the number of both pupae and adults is reduced during the development from eggs to adults. Data are displayed as mean ± SEM. (**B**) Pygo-F773W mutation reduces lifespan. Pygo-F773W mutant female and male flies showed median survivals of 12 and 15 days, and their respective control *w^1118^* female and male flies had median survivals of 57 and 51 days, respectively. This reduction was significant in both females and males (*** *p* < 0.001, Mandel–Cox log-rank test). Graph plots % survival (n = 200) versus time (in days) post-eclosion.

**Figure 8 ijms-25-05998-f008:**
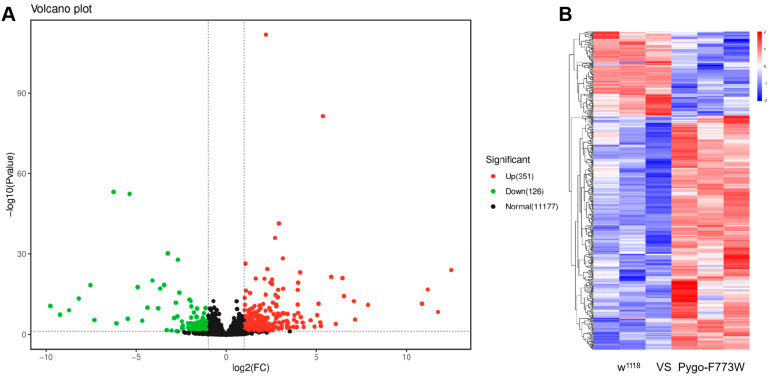
RNA-seq analysis reveals differential gene expression in 3-day-old PygoF773W mutant flies. (**A**) Volcano plot showing the log10 *p*-value and the log2 fold change of normalized counts among different genotypes. Each dot on the plot represents a single gene; green dots represent downregulated genes; red represent upregulated genes; black dots represent genes with no significant changes. Dashed lines indicate a *p*-value cutoff of 0.05 and a foldchange cutoff of 2. (**B**) Heatmap of DEGs, where rows and columns represent genes and samples, respectively. Red and blue indicate high and low expression levels, respectively. Darker colors denote more pronounced significant differences.

**Figure 9 ijms-25-05998-f009:**
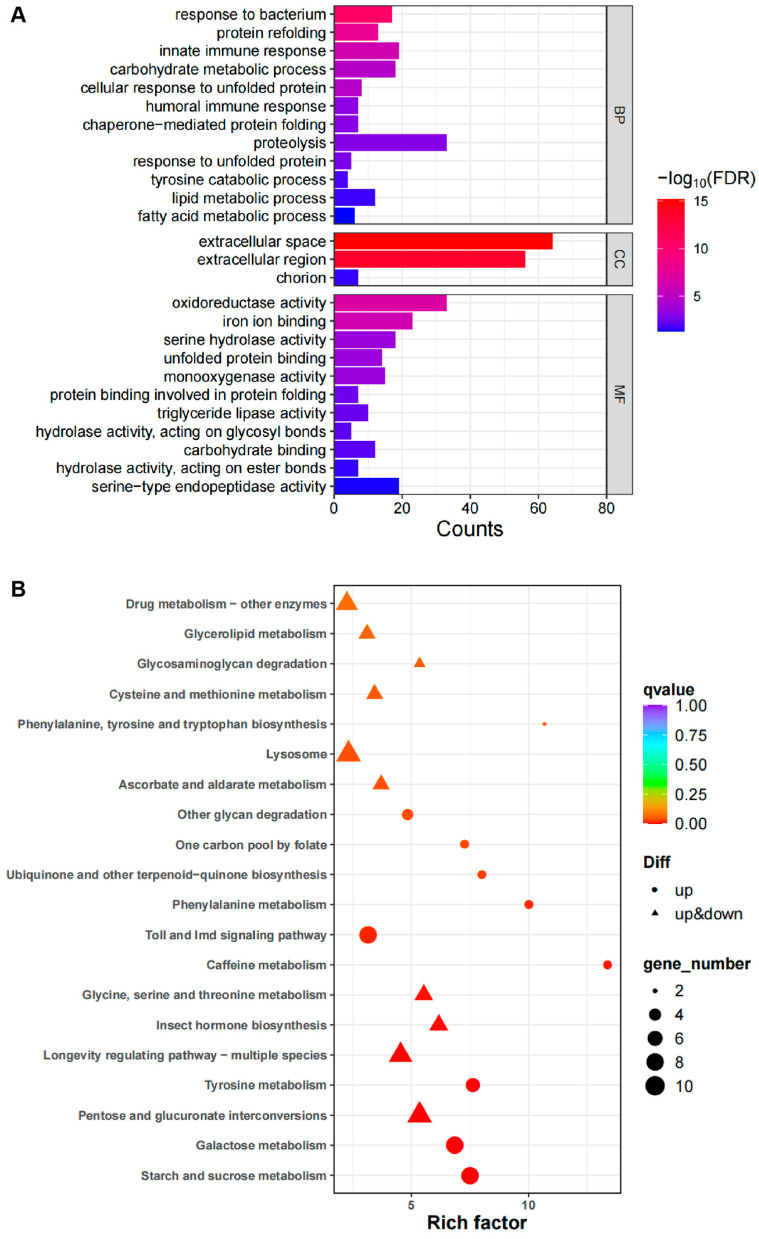
GO enrichment analysis and Kyoto Encyclopedia of Genes and Genomes (KEGG) pathway enrichment analysis of DEGs. (**A**) GO enrichment analysis; (**B**) KEGG pathway enrichment analysis. The figure shows the top 20 pathways with the smallest significant Q value.

**Figure 10 ijms-25-05998-f010:**
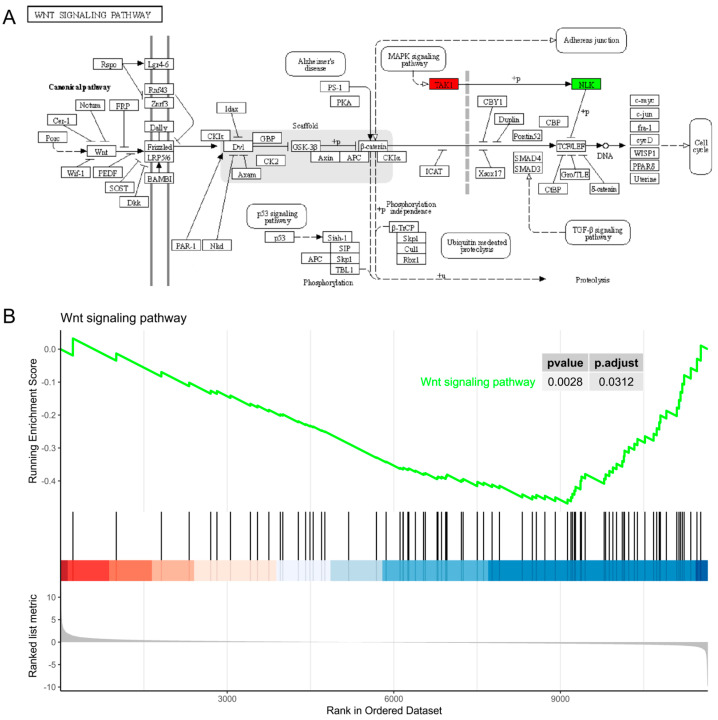
Differential genes analyzed in the Wnt signaling pathway. (**A**) Wnt signaling pathway annotation map of DEGs. The box represents the protein, red represents up-regulation, green represents down-regulation, and the rounded box represents another metabolic pathway. (**B**) GSEA analysis of all differentially expressed genes in the Wnt signaling pathway. The black vertical line in the abscissa represents the genes in the Wnt pathway; the ordinate in the above figure represents the dynamic enrichment score; the green curve represents the enrichment score of the gene set at each position; the peak of the green curve is the ES value of Wnt pathway. If the ES value is positive, the current function shows an upward trend; if the ES value is negative, the current function shows a downward trend.

**Figure 11 ijms-25-05998-f011:**
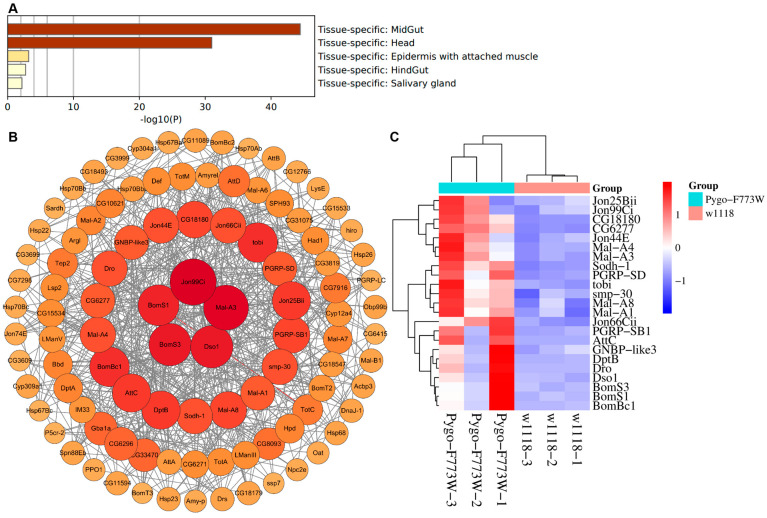
Tissue-specific enrichment analysis and core gene screening of DEGs. (**A**) Summary of enrichment analysis in PaGenBase. (**B**) Construction of the PPI network, only showing genes with degrees greater than 10. (**C**) Heatmap of 18 hub genes and 5 key genes based on RNA-seq data.

**Figure 12 ijms-25-05998-f012:**
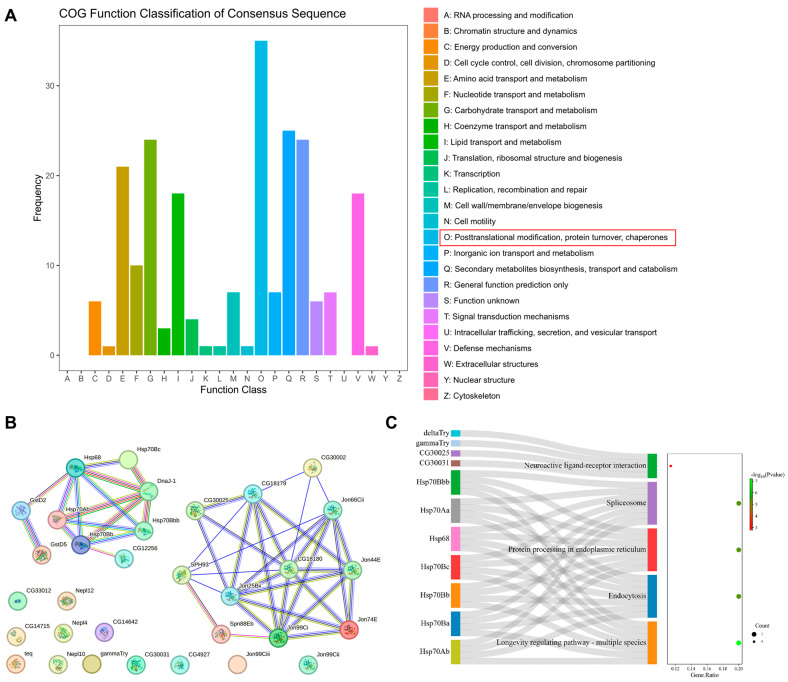
Pygo-F773W is related to posttranslational modification. (**A**) Cluster of orthologous groups of proteins (COG) analysis clustered the genes into 24 clusters. The abscissa represents the number of genes, and the ordinate represents the COG entry annotated. Red box represents the COG type with the largest number of differentially expressed genes. (**B**) PPI network analysis using STRING online tools. (**C**) KEGG pathway enrichment analysis of the DEGs related to posttranslational modification.

**Figure 13 ijms-25-05998-f013:**
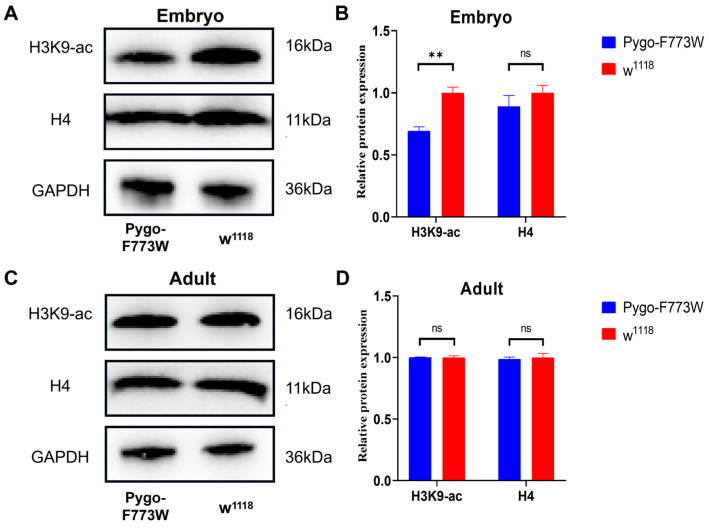
Western blot analysis. (**A**,**B**) Western blot and quantitative analysis of H4 and histone H3 lysine 9 acetylation (H3K9-ac) in Pygo-F773W-mutant and *w^1118^ Drosophila* embryos. (**C**,**D**) Western blot and quantitative analysis of H4 and histone H3 lysine 9 acetylation (H3K9-ac) in Pygo-F773W-mutant and *w^1118^* adult *Drosophila*. Data are displayed as mean ± SEM (n = 3). ns *p* > 0.05; ** *p* < 0.01 (unpaired *t*-test).

**Figure 14 ijms-25-05998-f014:**
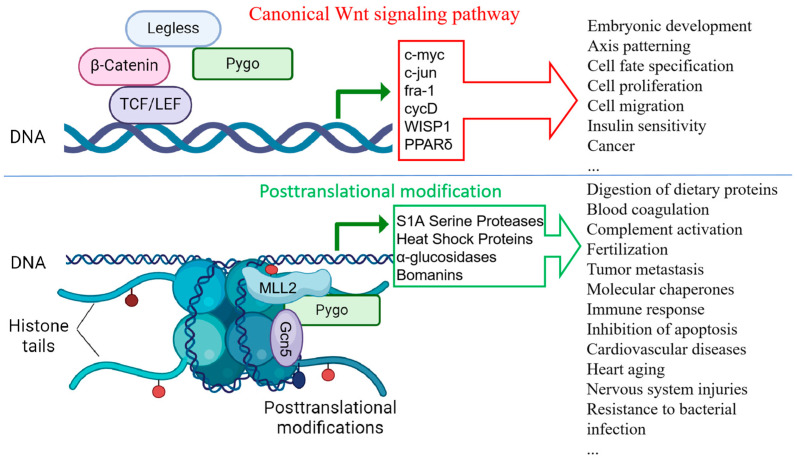
Pygo function diagram in canonical Wnt signaling pathway and posttranslational modification (created using BioRender.com).

## Data Availability

All data generated or analyzed during this study are included in this published article and its Appendix A.

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
