# Peer review of "Pygo-F773W Mutation Reveals Novel Functions beyond Wnt Signaling in Drosophila"

_ijms, 2024, doi:10.3390/ijms25115998_

Round 1
Reviewer 1 Report
Comments and Suggestions for Authors
The manuscript “Pygo-F773W Mutation Reveals Novel Functions Beyond Wnt 2 Signaling in Drosophila” raises a relevant question concerning the possible roles of Pygo independent of its participation in the Wingless signaling pathway. To analyze this aspect of Pygo function, the authors generate the mutation F773Win the pygo locus, making a protein that resemble in this position the human Pygo orthologs. This is supposed to increase the binding of the mutant protein to Histone H3 di- and trimethylated in lysine 4, but this expectation does not appear to hold true. The new mutant pygo F773W turns out to be viable in homozygosis, and the authors made a very comprehensible effort to characterize the developmental effects of this mutant on organism viability, fertility, mobility and gene expression in adult flies. There is by no means a clear way to connect the array of phenotypes observed in this mutant background with the possible normal functions of pygo acting independently of wingless signalling. This would require some comparison between this allele and other pygo alleles or RNAi approach. Without this set of data, it is not possible to identify functions of pygo altered by this mutation, and the authors should discuss this. A identification of pygo requirements in adult flies may be accomplished by expressing pygo RNAi in specific tissues such as cardiac cells, neurons, muscles, germ cells. This will allow at least to identify if any of the pygo F773W phenotypes is also observed upon a clean reduction of pygo expression in particular tissues.
In this manner, the phenotypes of pygo F773W, which clearly are not related to wingless signalling, are still a bit of a curiosity that is difficult to relate to normal functions of pygo. Still, this manuscript has value in the sense that it may represent an interesting starting point to further dissect pygo function. The authors identify that the greatest number of genes differentially expressed in pygo F773W correspond to posttranslational modification, protein turnover and chaperones. This makes me wonder to what extent the mutant cause some level of unfolded protein response, which may be the cause of the observed phenotypes.
Comments on the Quality of English LanguageThe quality of english is correct
Reviewer 2 Report
Comments and Suggestions for Authors
In this study, the authors generated the pygo-F773W mutant flies to analyze the molecular and biological functions of pygo using CRISPR/Cas9 technology. Using this mutant, the authors found that pygo is associated with reproduction, locomotion, heart function, and lifespan. Furthermore, they analyzed which genes are changed in expression due to the pygo-F773W mutation. Their data will provide much information for pygo study. However, there are some points the authors should further elucidate, and it seems that clearer data are needed to explain pygo functions.
Major points:
1. In this study, the authors generated a pygo-F773W mutant using CRISPR/Cas9 technology. To enhance the reader’s understanding, please provide the mutation site with domain information of pygo. Describe the reason why the authors generated and analyzed the pygo-F773W mutant. If this mutation information is from humans, please provide the mutation site with alignment results of pygo between Drosophila and humans.
2. In the Discussion section, the authors described that the phenotypes are similar between the pygo knockdown and the pygo-F773W mutation. Have the authors determined if the expression level of pygo was not altered in the pygo-F773W mutant compared with the control?
3. Is the expression pattern of pygo different across tissues or developmental stages?
4. How about the locomotor ability in the larval stage of the pygo-F773W mutant flies? Furthermore, is the significantly shortened post-eclosion lifespan associated with lower locomotor ability (flies may be easily stuck to the food) and/or expression changes of other genes due to the pygo-F773W mutation?
5. The authors presented the results of KEGG pathway analysis separately for up-regulated and down-regulated genes. What about the GO enrichment analysis results for both up and down-regulated genes?
6. In this study, some results (e.g., Wnt signaling and H3K9ac) are different from the expected results, as mentioned by the authors. Have the authors checked the possibility of other mutations induced by CRISPR/Cas9 technology?
7. It seems that there are conflicting results. The authors described it like this: “Overall, it was observed that a greater number of genes exhibited upregulation compared 246 to those showing downregulation following the Pygo-F773W mutation, indicating a prevalent trend towards enhanced transcriptional activity.” However, the H3K9ac levels were reduced in the embryo stage of pygo-F773W mutants or not changed in the adult stage of pygo-F773W mutants. How can we understand these results? Have the authors checked other histone markers?
8. In this study, the authors generated the Pygo-F773W mutants and analyzed their phenotypes. Then, the authors performed transcriptome analysis and western blotting to identify gene expression changes. However, it seems that the molecular results are not sufficient to support the phenotypes induced by the Pygo-F773W mutation.
Minor points:
1. Page 11 line 300-301: “Because it has been previously reported that the organs where Pygo acts in a Wnt 300 independent manner mainly include eye, teeth, testis, brain, intestine and salivary gland.” – please add the proper references to this sentence.
2. The labels on the figures are small, making it difficult to read them. Please increase the size of the labels on the figures.
Round 2
Reviewer 2 Report
Comments and Suggestions for Authors
The authors have addressed most of the reviewer's points in the revised manuscript, but some points still need clearer data. I have no further comments.